# Youku-mPLUG: A 10 Million Large-scale Chinese Video-Language Pre-training Dataset and Benchmarks

## Abstract

We firstly release the largest public Chinese high-quality video-language dataset named Youku-mPLUG, which is collected from Youku[1], a well-known Chinese video-sharing website, with strict criteria of safety, diversity, quality, and copyright. Youku-mPLUG contains 10 million Chinese video-text pairs filtered from 400 million raw videos across a wide range of 45 diverse categories for large-scale pre-training. In addition, to facilitate a comprehensive evaluation of video-language models, we carefully build the largest human-annotated Chinese benchmarks covering three popular video-language tasks across cross-modal retrieval, video captioning, and video category classification. We also provide comprehensive benchmark evaluations of models across different architectures including encoder-only (i.e., ALPRO), encoder-decoder (i.e., mPLUG-2), and decoder-only (i.e., mPLUG-Video) for comparison. Especially, we train the first Chinese Multimodal LLM with only 1.7% trainable parameters for video understanding. Experiments show that models pre-trained on Youku-mPLUG gain up to 23.1% improvement in video category classification. Besides, mPLUG-video achieves a new state-of-the-art result on these benchmarks with 80.5% top-1 accuracy in video category classification and 68.9 CIDEr score in video captioning, respectively. Finally, the 2.7B version of mPLUG-video demonstrates impressive instruction and video understanding ability. The zero-shot instruction understanding experiment indicates that pretraining with Youku-mPLUG can enhance the ability to comprehend overall and detailed visual semantics, recognize scene text, and leverage open-domain knowledge.

## 1 Introduction

With the release of large-scale English video-language datasets (e.g., Howto100M(Miech et al., 2019) and WebVid-2.5M(Bain et al., 2021)), video-language pre-training (VLP) has achieved the superior performance on various downstream tasks, such as video-text retrieval, video question answering, and video captioning. Moreover, the recent multimodal LLM in video (e.g., VideoChat(Li et al., 2023b), Flamingo(Alayrac et al., 2022)) has demonstrated strong zero-shot video understanding ability based on these large-scale datasets. Compared with the English VLP community as Tab. 1, the lack of large-scale and high-quality public Chinese VLP datasets hinders the research of Chinese video-language pretraining and multimodal LLM. In addition, publicly available benchmarks as Tab. 2 are also missing for the Chinese VLP community. These limitations will result in two significant issues. Firstly, the development and application of Chinese VLP and multimodal LLM are being lagged behind. Secondly, the comparison between different methods becomes challenging due to the fairness issue that some works are able to achieve surprisingly good performance by using secret downstream benchmarks. While some methods translate English text into Chinese (Madasu et al., 2022) or annotate the dataset based on the English video (Wang et al., 2019), there remains an intrinsic linguistic and cultural gap between English and Chinese.

To facilitate the research and application of Chinese VLP, we release the first and largest public Chinese video-language pretraining dataset and benchmarks named Youku-mPLUG, which is collected from Youku, a well-known Chinese video-sharing website with strict criteria of safety, diversity,

---

[1]https://www.youku.com

Table 1: Statistics of Youku-mPLUG and its comparison with existing video-language pre-training datasets.

| Dataset Name | Language | # Videos | # Text | Avg. Len (secs) | Duration (hrs) | Domain | Availability |
|---|---|---|---|---|---|---|---|
| HowTo100M (Miech et al., 2019) | English | 136M | 136M | 3.6 | 135K | Instruction | ✓ |
| YT-Temporal-180M (Zellers et al., 2021) | English | 180M | 180M | - | - | Instruction | ✓ |
| HD-VILA-100M (Xue et al., 2022) | English | 103M | 103M | 13.4 | 372K | Open | ✓ |
| WebVid10M (Bain et al., 2021) | English | 10M | 10M | 18.0 | 52K | Open | ✓ |
| ALIVOL-10M (Lei et al., 2021a) | Chinese | 103M | 110M | 34.6 | 99K | E-Commerce | ✗ |
| Kwai-SVC-11M (Nie et al., 2022b) | Chinese | 11M | 4M | 57.9 | 177K | Open | ✗ |
| CREATE-10M (Zhang et al., 2022) | Chinese | 10M | 10M | 29.8 | 83K | Open | ✗ |
| CNVid-3.5M (Gan et al., 2023) | Chinese | 3.5M | 3.5M | 36.2 | 35K | Open | ✗ |
| **Youku-mPLUG** | Chinese | 10M | 10M | 54.2 | 150K | Open | ✓ |

Table 2: Statistics of Youku-mPLUG and its comparison with existing video-language downstream datasets.

| Dataset Name | Language | # Sample | Domain | Retrieval | Classification | Caption | Availability |
|---|---|---|---|---|---|---|---|
| MSRVTT (Xu et al., 2016) | English | 10K | Open | ✓ | ✓ | ✓ | ✓ |
| DiDeMo (Anne Hendricks et al., 2017) | English | 27K | Flickr | ✓ | ✗ | ✗ | ✗ |
| MSVD (Chen & Dolan, 2011) | English | 10K | Open | ✓ | ✓ | ✓ | ✓ |
| LSMDC (Rohrbach et al., 2015) | English | 118K | Movie | ✓ | ✓ | ✗ | ✓ |
| ActivityNet (Krishna et al., 2017) | English | 100K | Open | ✓ | ✓ | ✗ | ✓ |
| VATEX (Wang et al., 2019) | English/Chinese | 41K | Kinetics-600 | ✓ | ✗ | ✓ | ✓ |
| BFVD (Zhang et al., 2020) | Chinese | 43K | E-Commerce | ✓ | ✗ | ✗ | ✗ |
| FFVD (Zhang et al., 2020) | Chinese | 32K | E-Commerce | ✓ | ✗ | ✗ | ✗ |
| CREATE-210K (Zhang et al., 2022) | Chinese | 216K | Open | ✓ | ✗ | ✓ | ✗ |
| **Youku-mPLUG** | Chinese | 365K | Open | ✓ | ✓ | ✓ | ✓ |

quality and copyright. Youku-mPLUG contains 10 million video-text pairs for pre-training and 0.3 million videos for downstream benchmarks. For the pre-training dataset, we collect 10 million high-quality video-text pairs filtered from 400 million raw videos with the strict criteria of safety, diversity, and quality. **Safety**, the dataset is subject to heavy filtering and restrictions through an in-house multi-level risk detection system to prevent any content related to high risks; **Diversity**, the videos are carefully classified into 45 diverse categories covering various domains, e.g., Daily life, Comedy, and Pet, with a balanced distribution; **Quality**, we have conducted strict data cleaning at both the text and video levels, while using Chinese image-text pre-trained model to improve the data quality. Furthermore, We build the largest human-annotated Chinese benchmarks covering Cross-modal Retrieval, Video Captioning, and Video Category Classification for comprehensive evaluation of video-language models and downstream applications. For each downstream task, we hire well-educated people and adopt a two-step verification to ensure the quality and diversity of the annotations In concrete, We would first hire a group of well-educated people to annotate a small fraction of data with provided annotation details and instructions. Then we scrutinize the annotated data and filter out those annotators who have extremely poor annotation quality. We also revised the annotation instructions according to the problems during the first-round annotation. After that, we give another small fraction of the data for annotation. If the quality of these annotations meets the requirement, we would provide all of the data for labeling. Otherwise, we repeat the previous checking procedure.

Besides, we investigate popular video-language models, the encoder-only model ALPRO (Li et al., 2022b) and the encoder-decoder model mPLUG-2 (Xu et al., 2023) pre-trained on Youku-mPLUG. Drawing inspiration from the idea of modularization (Li et al., 2022a; Xu et al., 2023; Ye et al., 2023), we propose the modularized decoder-only model mPLUG-video with limited trainable parameters, which consists of the trainable video encoder, visual abstractor module, and the frozen pre-trained LLM decoder. We first obtain dense video representations from the video encoder. Then, we employ the visual abstractor module to summarize visual information with several learnable tokens. Finally, the visual representations are combined with text queries and fed into the frozen LLM decoder to generate the response. Experiments show that models pre-trained on Youku-mPLUG gain up to 23.1% improvement in video category classification. With the proposed dataset, mPLUG-video achieves 80.5% top-1 accuracy in video category classification and 68.9 CIDEr score in video captioning, respectively. It becomes new state-of-the-art results on these benchmarks. Moreover, we scale up mPLUG-video based on frozen Bloomz(Workshop et al., 2023) as Chinese multimodal LLM with only 1.7% trainable parameters, which demonstrates impressive instruction and video

understanding ability. As an insight, our zero-short video instruction understanding test validates that Youku-mPLUG can strengthen the scene text recognizing ability and incorporate open-domain knowledge for video understanding. Qualitative results can be found in the Supplementary Material. These pre-trained models have also been released to facilitate the research and application of Chinese video-language pre-training.

In summary, our main contributions are:

- We release the first and largest Chinese video-language pretraining dataset and benchmarks named Youku-mPLUG.

- We provide comprehensive benchmark evaluations of models across different architectures including encoder-only (i.e., ALPRO), encoder-decoder (i.e., mPLUG-2), and our proposed modularized decoder-only mPLUG-video pre-trained on Youku-mPLUG for comparison.

- We scale up and release mPLUG-video based on Bloomz as Chinese multimodal LLM with only 1.7% trainable parameters, which demonstrates the impressive zero-shot instruction and video understanding ability.

- Experiments show that models pre-trained on Youku-mPLUG gain a significant improvement over baselines and mPLUG-video achieves state-of-the-art results on these benchmarks.

## 2 RELATED WORK

**Video-Language Pre-training Datasets** Large-scale datasets have proven effective for video-language representation learning. Previously, most video-language models were trained on the HowTo100M dataset (Miech et al., 2019), which comprises 136 million video clips from 1.22 million instructional YouTube videos. However, this dataset is limited to the instructional domain and is unsuitable for generalization. To overcome this constraint, Zeller et al. (Zellers et al., 2021) and Xue et al. (Xue et al., 2022) propose the YT-Temporal-180M and HD-VILA-100M corpus, respectively. Meanwhile, to reduce the noise in subtitles, Bain et al. (Bain et al., 2021) introduce the Webvid10M dataset which is inspired by the collection schemes of Conceptual Caption datasets (Sharma et al., 2018). However, these datasets are limited to English language corpus and cannot be directly applied to the Chinese domain. Although there exist some large-scale Chinese video-language datasets such as ALIVOL (Lei et al., 2021a), Kwai-SVC (Nie et al., 2022a), CREATE-10M (Zhang et al., 2022), and CNVid-3.5M (Gan et al., 2023), none of them have been publicly released to date, which hinders the progress of research in the Chinese video-language learning field. To address this gap, we present Youku-mPLUG, the largest Chinese high-quality video-language dataset, to facilitate future research on large-scale video-language learning in the Chinese language.

**Video-Language Downstream Benchmarks** For evaluating video-language pre-training models, researchers have proposed several downstream tasks such as video-text retrieval, video question answering, and video captioning for performance evaluation. For instance, MSRVTT (Xu et al., 2016), DiDeMo (Anne Hendricks et al., 2017), and LSMDC (Rohrbach et al., 2015) are commonly adopted for text-video retrieval evaluation. Similarly, MSRVTT-QA (Xu et al., 2017), MSVD-QA (Xu et al., 2017), and T-GIF (Jang et al., 2017) are widely used for video question evaluation. Meanwhile, MSRVTT-Caption (Xu et al., 2016) and MSVD-Caption (Chen & Dolan, 2011) are commonly used for video caption evaluation. However, these datasets are primarily collected from YouTube, which is not entirely suitable for the Chinese domain. Furthermore, while there are some Chinese benchmark datasets such as CREATE (Zhang et al., 2022) and VATEX (Wang et al., 2019), they are not fully released and only evaluate one aspect of the model's performance. Additionally, there is a lack of systematic video language downstream benchmarks or leaderboards for Chinese video-language pre-training evaluation. Consequently, we propose three downstream benchmarks, including video category classification, video-text retrieval, and video captioning, for evaluating models' performance on Youku-mPLUG. These benchmarks are specifically designed for the Chinese domain and are intended to fill the gap in existing English benchmarks, which may not be entirely suitable for Chinese video-language pre-training evaluation.

**Video-Language Pre-training Models** In recent years, there has been a growing interest in video-language pre-training, and various methods have been proposed to explore this area. Traditional approaches (Luo et al., 2020; Li et al., 2020) rely on pre-extracted, dense video frame or clip features for video-language representation. In contrast, ClipBERT (Lei et al., 2021b) introduces a sparse sampling strategy that facilitates end-to-end learning while simultaneously improving performance.

Building upon this strategy, many approaches (Bain et al., 2021; Ge et al., 2022) have been developed, which incorporate novel architectures and pre-training tasks for video-language learning. For example, Frozen (Bain et al., 2021) and BridgeFormer (Ge et al., 2022) employ contrastive learning to align the semantics of paired video and text in the same embedding space. Additionally, ALPRO (Li et al., 2022b), TW-BERT (Yang et al., 2023), mPLUG-2 (Xu et al., 2023), and HiTeA (Ye et al., 2022) fuse video and language features to generate video-language representations for understanding and generation. Recently, large language models such as GPT-3 (Brown et al., 2020), Bloom (Workshop et al., 2023), and LLaMA (Touvron et al., 2023) have demonstrated significant zero-shot generalization abilities, which are advantageous for the vision-language field. For instance, BLIP-2 (Li et al., 2023a), miniGPT-4 (Zhu et al., 2023), and mPLUG-Owl (Ye et al., 2023) exhibit robust zero-shot generalization and conversation capabilities by aligning vision and language models. In this work, we provide a decoder-only video-language model mPLUG-video pre-trained on our Youku-mPLUG dataset with a strong generalization performance in terms of both video-language understanding and generation.

## 3 YOUKU-MPLUG DATASET CREATION

To fill in the blank of the public Chinese video-text pre-training dataset and benchmarks, we release the largest public Chinese Video-language dataset named Youku-mPLUG collected with the strict criteria of safety, diversity, and quality from Youku, a Chinese video-sharing website. Youku-mPLUG contains 10 million video-text pairs for pre-training and 0.3 million videos for downstream benchmarks covering Video-Text Retrieval, Video Captioning, and Video Category Classification. Randomly sampled examples are shown in Figure 1.

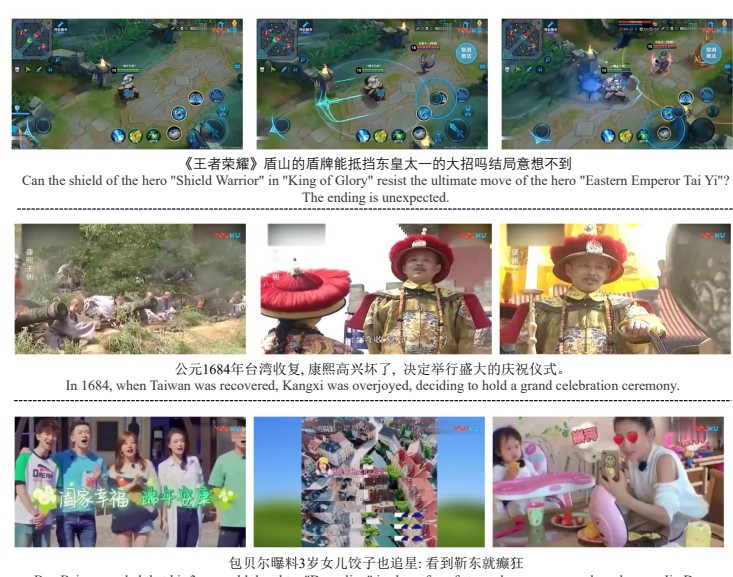

《王者荣耀》盾山的盾牌能抵挡东皇太一的大招吗结局意想不到
Can the shield of the hero "Shield Warrior" in "King of Glory" resist the ultimate move of the hero "Eastern Emperor Tai Yi"? The ending is unexpected.

公元1684年台湾收复，康熙高兴坏了，决定举行盛大的庆祝仪式。
In 1684, when Taiwan was recovered, Kangxi was overjoyed, deciding to hold a grand celebration ceremony.

包贝尔曝料3岁女儿饺子也追星: 看到靳东就癫狂
Bao Beier revealed that his 3-year-old daughter "Dumpling" is also a fan of stars: she goes crazy when she sees Jin Dong.

Figure 1: Random sampled examples in Youku-mPLUG.

### 3.1 PRE-TRAINING DATASET CONSTRUCTION

For the pre-training dataset, we filter 10 million high-quality video-text pairs from 400 million raw videos with strict safety, diversity, and quality criteria. In terms of safety, the dataset is heavily filtered and restricted by an internal multi-level risk detection system with both multimodal model detection and manual review processes to prevent any content related to pornography, violence, terrorism, discrimination, abuse, or other high risks. In specific, the safety detection system primarily consists of two components. Firstly, we utilize in-house visual and language models to identify potentially hazardous content in videos and title information, including pornography, violence, terrorism, discrimination, abuse, etc., and ensemble the results. Secondly, a crowd-sourcing platform is employed for manual re-checking, in cases where it is challenging for the models to differentiate (e.g., when scores are indistinguishable). The annotation results will be fed to the model for more refined training. Regarding diversity, we have applied video fingerprinting technology to eliminate videos that are completely identical. With the hierarchical multi-label classification model (Giunchiglia & Lukasiewicz, 2020), the videos are carefully classified into 20 super categories and

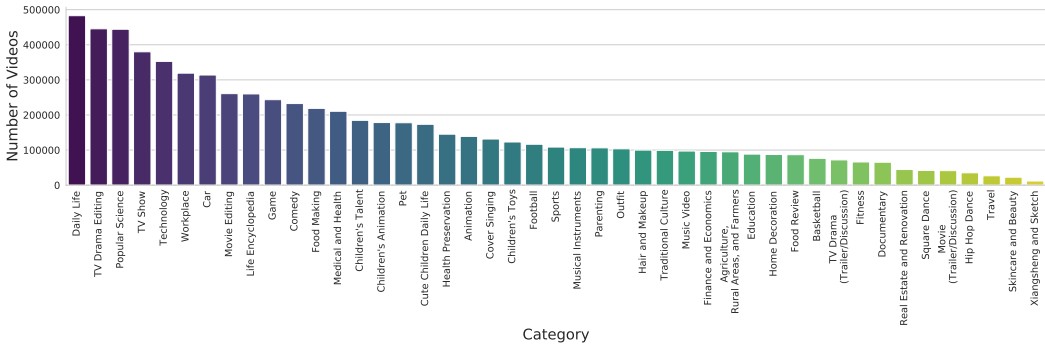

Figure 2: The distribution of the number of videos in each common category.

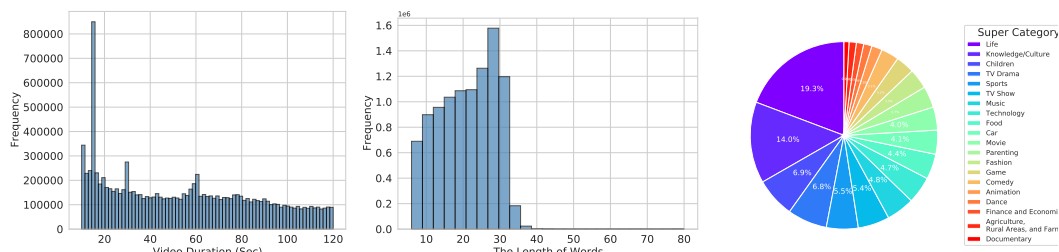

Figure 3: Youku-mPLUG dataset statistics: we report the histogram of video duration in seconds (left), the histogram of title length in words (middle), and the ratios of the categories in each super-category (right).

45 common categories as Fig. 2, covering various domains, with a balanced distribution. To ensure high quality, we have conducted strict data cleaning at both the text and video levels. For text, we have imposed language restrictions on video titles, requiring the length to be between 5 and 30 words and including at least 5 Chinese characters while filtering out those with obvious advertising or meaningless content. In terms of video quality and integrity, we have specifically chosen recently uploaded videos with durations ranging from 10 to 120 seconds to ensure clear and complete content. Further, we also employ the Chinese image-text pre-trained model CLIP (Yang et al., 2022) to improve the data quality by deprecating those with low similarities between the mean frame features and text features. Fig. 3 shows the statistics of video duration and word length. Furthermore, to safeguard the copyright of videos, we manually insert a 2-second shallow watermark at the start of each video, which is indispensable to open-source these videos. As demonstrated in (Bain et al., 2021), these watermarks do not impact the performance of the model.

## 3.2 DOWNSTREAM BENCHMARK CONSTRUCTION

For the downstream benchmark, we design three types of tasks including video-text retrieval, video category classification, and video captioning to evaluate the performance in terms of understanding and generation. The statistics of these three different datasets are summarized in Tab. 3.

Table 3: Statistics of Youku-mPLUG benchmark datasets. # pairs indicates the number of video-text pairs.

| Task | Train (# Pairs) | Val (# Pairs) | Test (# Pairs) |
|------|-----------------|---------------|----------------|
| Video Category Classification | 100,023 | 14,678 | 20,026 |
| Video-Text Retrieval | 37,595 | 7,271 | 7,414 |
| Video Captioning | 170,866 | 7,510 | 7,705 |

**Video Category Classification** Our initial step involves randomly selecting a substantial number of videos based on category frequency. Next, we collect the video categories from the Youku database, which are auto-generated by an online model. It is important to note that this model's accuracy is approximately 94% when considering historical prediction data, thus not entirely reliable.

Consequently, we put forth additional efforts to ensure the quality and accuracy of our datasets by manually verifying each video and its corresponding title in the benchmark datasets. Prior to annotation, we supply a smaller dataset containing 100 videos, along with their metadata, including titles and categories generated by the online prediction model. Annotators are then tasked with confirming the assigned categories in relation to the videos and their titles. They must also assign a relevance score, which ranges from 1 to 5. A higher score suggests a greater likelihood of the video belonging to the given category, and those with scores above 3 are retained. Annotators with error rates exceeding 2.5% are disqualified. After eliminating unsuitable annotators, we proceed with annotating the video category classification dataset. To ensure the utmost accuracy, particularly for the validation and testing sets, we engage three annotators to verify each video.

**Video Captioning** The video captioning task requires the model to generate a concise sentence describing a video clip's content and title. To create the dataset, we randomly sample around 80,000 videos based on category frequency distribution and employ a color histogram-based approach for segmenting each video into shots (Mei et al., 2014). To ensure an accurate understanding of the video content and produce precise descriptions, we engage several annotators who are native Chinese speakers with strong educational backgrounds. As part of the pre-annotation process, we assign 25 random videos to each annotator, requesting them to create captions that include the subject and object in the video, as well as relevant descriptions of actions and background. The captions must consist of at least 15 Chinese characters. Following the pre-annotation stage, annotators proceed with annotating the datasets and split them into the training, validation, and testing sets. Especially, to prevent data leakage, clips from the same video or sharing the same title are exclusively assigned to either the training or testing sets. Moreover, for the validation and testing datasets, we enlist more than three individuals to annotate the video clips, promoting diversity and quality.

**Video-Text Retrieval** Similar to the annotation procedures video captioning task, we first segment the video into clips using a color histogram-based method. Then, these video clips are assigned to different native Chinese speakers for labeling the clips. We also adopt the two-step verification procedure in which each collected description must be reviewed. In addition, we ensure that clips from the same video or those with identical text titles are not exclusively included in the training or test set to prevent potential data leakage.

## 4 METHODOLOGY

Since the pre-trained large language model shows incredible zero-shot and generalization abilities on various tasks, we use the off-the-shelf Chinese large language model (e.g. GPT-3 (Brown et al., 2020)) for efficient modularized training. To this end, we propose mPLUG-video, a decoder-only based video-language model that leverages the frozen large language model. Specifically, our model consists of a video encoder, a visual abstractor module, and a language decoder, as illustrated in Figure 4. Besides, we only train the video encoder and visual abstractor containing limited parameters , which reduces the computation burden significantly.

### 4.1 ARCHITECTURE

**The Video Encoder** We leverage a 12-layer TimeSformer (Bertasius et al., 2021) to extract the video features, with $224 \times 224$ input frames. We sparsely sample $T$ frames from each video $\mathcal{V}$, where the TimeSformer first divides the video frames into $N$ non-overlapping patches and flattens them into a sequence of $T \times N$ patches. Then these patches are fed into the patch projection layers for patch representation. To encode the position of each patch, we add learnable embeddings to encode each patch's spatial and temporal position. Then the TimeSformer applies divided spatiotemporal attention to yield video representation $V \in \mathbb{R}^{(T \times N) \times D}$, where $D$ is the hidden dimension of the video representation.

**Visual Abstractor Module** To mitigate the computation burden with the lengthy video sequences, we introduce visual abstractor module which utilizes learnable queries $Q \in \mathbb{R}^{M \times D}$ for reducing the length of video sequence as follows:

$$\tilde{Q} = CrossAttention(Q, V, V), \tag{1}$$

$$\tilde{Q} = FFN(\tilde{Q}) + \tilde{Q}, \tag{2}$$

where $CrossAttention(x, y, z)$ is the cross-attention layer with Query $x$, Key $y$, and Value $z$. The $FFN(\cdot)$ is the feed-forward layer (Vaswani et al., 2017). Finally, we obtain the reduced video sequence $\tilde{Q} \in \mathbb{R}^{M \times D}$.

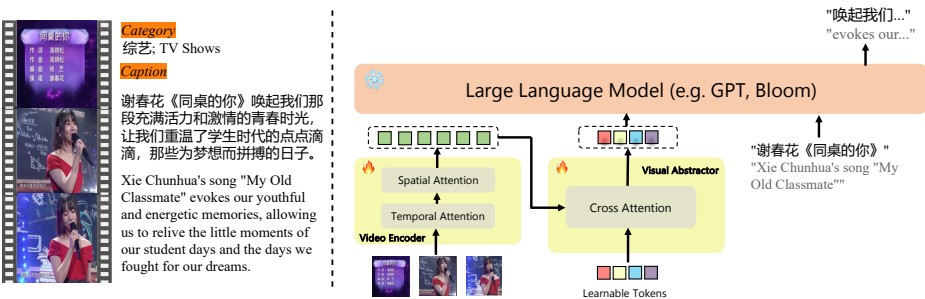

Figure 4: The overview of mPLUG-video.

**The Language Decoder** Since pre-trained large language models demonstrate strong zero-shot capabilities in text generation, we utilize them as the general text decoder for multi-modal inputs while keeping it frozen. In specific, we treat the video as a foreign language and concatenate the reduced video sequence with the text token features obtained from the text embedding layer. Then, the video and text token features are jointly fed into the large language model which is frozen for obtaining the video-guided language features. Finally, the video-guided language features are predicted for text tokens.

**Training Objective** We train mPLUG-video within an auto-regressive manner and adopt the next token prediction task for training. In detail, the model needs to complete the texts based on the given video, and the language modeling loss is calculated as:

$$\mathcal{L} = -\mathbb{E}_{(\mathcal{W},\mathcal{V})} \left[ \sum_{l=1}^{L} \log p(w_l | \mathcal{W}_{[0,l)}, \mathcal{V}) \right], \tag{3}$$

where $L$ denotes the total number of words in the text, and $\mathcal{W}$ denotes the word tokens.

## 4.2 APPLICATION TO DOWNSTREAM TASKS

**Video Captioning** Video captioning is considered an auto-regressive task. During the process of fine-tuning a video captioning dataset, the training objective remains the same as pre-training.

**Video Category Classification** We treat video category classification as a video caption task. Annotated category names of videos are regarded as ground-truth captions. We evaluate the accuracy of predictions based on whether the predicted category name exactly matches the ground-truth.

**Video-Text Retrieval** In contrast to mPLUG-2, which includes a contrastive head and a matching head for the retrieval task, our mPLUG-video cannot be directly used for retrieval tasks. Therefore, we input video-text pairs into the model and extract the feature of the last token. We obtain the matching score by applying an extra linear layer to the feature of the last token.

## 5 EXPERIMENTS

### 5.1 IMPLEMENTATION DETAILS

mPLUG-video leverages the pre-trained Chinese GPT-3 [2] [3] as the language decoder, and the video encoder is pre-trained on ImageNet (Ridnik et al., 2021). During pre-training, we sparsely sample 8 frames from each video preserving their order in-between, and resize them to $224 \times 224$. We use a batch size of 512 and train mPLUG-video for 10 epochs. We adopt the AdamW optimizer with $\beta = (0.9, 0.98)$, and set the learning rate and weight decay to 1e-4 and 1e-3 respectively. We warm up the training with 2000 warm-up steps then decay the learning rate with the cosine schedule. For downstream tasks, we use a batch size of 128 and train mPLUG-video for 10 epochs with a learning rate of 2e-5.

---

[2]https://modelscope.cn/models/damo/nlp_gpt3_text-generation_1.3B/summary
[3]https://modelscope.cn/models/damo/nlp_gpt3_text-generation_2.7B/summary

## 5.2 EVALUATION ON DOWNSTREAM TASKS

In this subsection, we evaluate the performance of ALPRO, mPLUG-2, and mPLUG-video on video category classification, video captioning, and video-text retrieval, respectively.

**Evaluation on Video Category Classification** We assess the performance of ALPRO, mPLUG-2, and mPLUG-video on video category classification tasks. We measure the top-1 and top-5 accuracy of each model. For the generation models, a generated category name that is exactly the same as ground truth can be regarded as a correct prediction. The comparison results are shown in Table 4. Our results reveal that mPLUG-video achieves the highest accuracy, with a top-1 accuracy of 80.57% and a top-5 accuracy of 98.15%. Interestingly, mPLUG-video (2.7B) outperforms mPLUG-video (1.3B), highlighting the importance of natural language understanding with a larger LLM decoder. Besides, mPLUG-video outperforms the other two models by utilizing the internal knowledge within LLM, showing the effectiveness of decoder-only architecture.

**Evaluation on Video Caption** We present in Table 4 the performance of models on Video Caption. ALPRO does not have a decoder module. Therefore, its performance was not reported. The performance of mPLUG-Video and mPLUG-2 are compared based on various metrics, including METEOR, ROUGE, CIDEr, and BLEU-4. It is found that mPLUG-video (2.7B) achieves higher scores than mPLUG-Video (1.3B) across all four metrics. Additionally, mPLUG-video obtains higher scores than mPLUG-2 on BLEU-4. These results suggest that pre-trained language models are essential and video captioning tasks based on our dataset are still challenging for existing methods We also present the results on VATEX (Wang et al., 2019) dataset in Table 5, which demonstrates models can benefit from pre-training on Youku-mPLUG.

**Evaluation on Video-Text Retrieval** Table 6 presents the performance comparison between models on video retrieval task. We observe that mPLUG-2 outperforms ALPRO, possibly due to the incorporation of universal layers that remove modality differences and generate superior uni-modal representations. We also notice that mPLUG-video performs poorly on video retrieval task. Since we only adopt language modeling as the pre-training task, it does not explicitly contain the video-language alignment with contrastive learning.

Table 4: Comparison results on Youku-mPLUG. Video category prediction and video captioning, respectively. For video category prediction, top-1 and top-5 accuracy are reported. For video captioning, we report BELU-4, METEOR, ROUGE, and CIDEr. * denotes the language model is frozen.

| Model | Video Category Prediction | | Video Captioning | | | |
| --- | --- | --- | --- | --- | --- | --- |
| | Top-1 Acc.(%) | Top-5 Acc.(%) | BLEU-4 | METEOR | ROUGE | CIDEr |
| ALPRO | 78.15 | 95.15 | - | - | - | - |
| mPLUG-2 | 77.79 | 92.44 | 43.7 | **27.6** | 52.9 | 67.7 |
| mPLUG-Video (1.3B)* | 80.04 | 98.06 | 46.4 | 26.5 | 52.9 | 67.7 |
| mPLUG-Video (2.7B)* | **80.57** | **98.15** | **47.1** | 26.7 | **53.3** | **68.9** |

Table 5: Comparison of results on VATEX of Video Captioning.

| Model | BLEU-4 | METEOR | ROUGE | CIDEr |
| --- | --- | --- | --- | --- |
| mPLUG-2 | 53.6 | 31.0 | 59.9 | 87.0 |
| mPLUG-Video (1.3B w/o pre-train)* | 49.2 | 29.4 | 58.1 | 76.8 |
| mPLUG-Video (1.3B w/ pre-train)* | **57.4** | **31.6** | **62.2** | **97.2** |

## 5.3 ABLATION STUDY ON MODALITIES

In this section, we investigate the contributions of different modalities to video-language modeling by leveraging the category classification task on our Youku-mPLUG. Table 7 presents the performance of the baseline model (ALPRO) trained with data of different modalities. Vision Modality and Language Modality denote the model trained with the corresponding modality of data (video frames or video captions). Youku-mPLUG Pre-Trained refers to the model pre-trained on Youku-mPLUG before fine-tuning. The results show that the performance of the model trained with the visual modality

Table 6: Comparison results on Youku-mPLUG. Video retrieval. We evaluate models on video retrieval (V2T) and text retrieval (T2V). we report the average of R@1, R@5 and R@10. * denotes the language model is frozen.

| Model | Video Retrieval | | | | | |
|---|---|---|---|---|---|---|
| | V2T | | | T2V | | |
| | R@1 | R@2 | R@10 | R@1 | R@5 | R@10 |
| ALPRO | 27.00 | 53.33 | 64.09 | 26.63 | 53.20 | 64.43 |
| mPLUG-2 | **38.45** | **65.48** | **75.18** | **38.45** | **65.48** | **75.18** |
| mPLUG-Video (1.3B)* | 7.01 | 20.33 | 29.67 | 7.01 | 20.33 | 29.67 |
| mPLUG-Video (2.7B)* | 7.62 | 21.24 | 31.39 | 7.62 | 21.24 | 31.39 |

Table 7: Comparison of different modalities and Youku-mPLUG on category classification task.

| Vision Modality | Language Modality | Youku-mPLUG Pre-Trained | Top-1 Acc.(%) | Top-5 Acc.(%) |
|---|---|---|---|---|
| ✓ | ✗ | ✗ | 63.51 | 89.89 |
| ✗ | ✓ | ✗ | 59.31 | 86.31 |
| ✓ | ✓ | ✗ | 69.40 | 90.07 |
| ✓ | ✓ | ✓ | **78.15** | **95.15** |

is higher than that with the language modality. This suggests that high-level language modalities may lose fine-grained visual clues, leading to failure in classification. Additionally, we observe that the model trained with both vision and language modalities achieves higher performance than unimodal models. This observation demonstrates the importance of modality complementarity in video understanding. Pre-training the model with Youku-mPLUG leads to a significant improvement in performance, emphasizing the importance of our Youku-mPLUG.

### 5.4 HUMAN EVALUATION OF ZERO-SHOT VIDEO INSTRUCTION UNDERSTANDING

To test the video instruction understanding ability of different models, we manually set 65 instructions based on 50 randomly-sampled videos (45 from Youku-mPLUG, 5 from HD-VILA-100M (Xue et al., 2022)). We compare the instruction understanding performance of three models: VideoLLaMA(Zhang et al., 2023), mPLUG-Video w/o pretrain and mPLUG-Video. VideoLLaMA is trained with visual instruction data from MiniGPT-4(Zhu et al., 2023), LLaVa (Liu et al., 2023) and Video-Chat (Li et al., 2023b), while the latter two models only utilize visual training data from LLaVa (Liu et al., 2023). We ask human annotators to score the models' responses. Following Self-Instruct(Wang et al., 2022), human annotators are asked to rate the response into four levels, where A means 'correct and satisfying response',

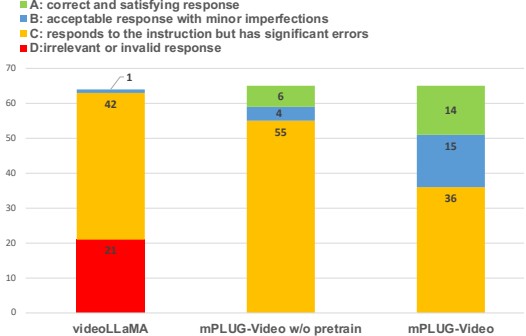

Figure 5: Human evaluation about zero-shot video instruction understanding on 65 cases.

B means 'acceptable response with minor imperfections', C means 'response to the instruction but has significant errors' and D means 'irrelevant or invalid response'. As shown in Fig. 5, with the pertaining on Youku-mPLUG, mPLUG-video achieves much better video instruction understanding and responding ability, demonstrating the effectiveness of our proposed pretraining data. Qualitative results can be found in the supplementary material.

## 6 CONCLUSION

In this paper, we introduce the largest high-quality video-language dataset in Chinese, called Youku-mPLUG. Additionally, we present a human-annotated benchmark that comprises three downstream tasks, i.e., Video-Text Retrieval, Video Captioning, and Video Category Classification. We propose a decoder-only model, mPLUG-video, that is modularized and pre-trained on Youku-mPLUG. Results from our experiments indicate that our evaluation set can effectively evaluate the video language comprehension and modeling abilities of models. Furthermore, pre-training on Youku-mPLUG leads to significant improvements, and our mPLUG-video achieves a new state-of-the-art performance.

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

## A QUALITATIVE RESULTS OF ZERO-SHOT VIDEO INSTRUCTION UNDERSTANDING

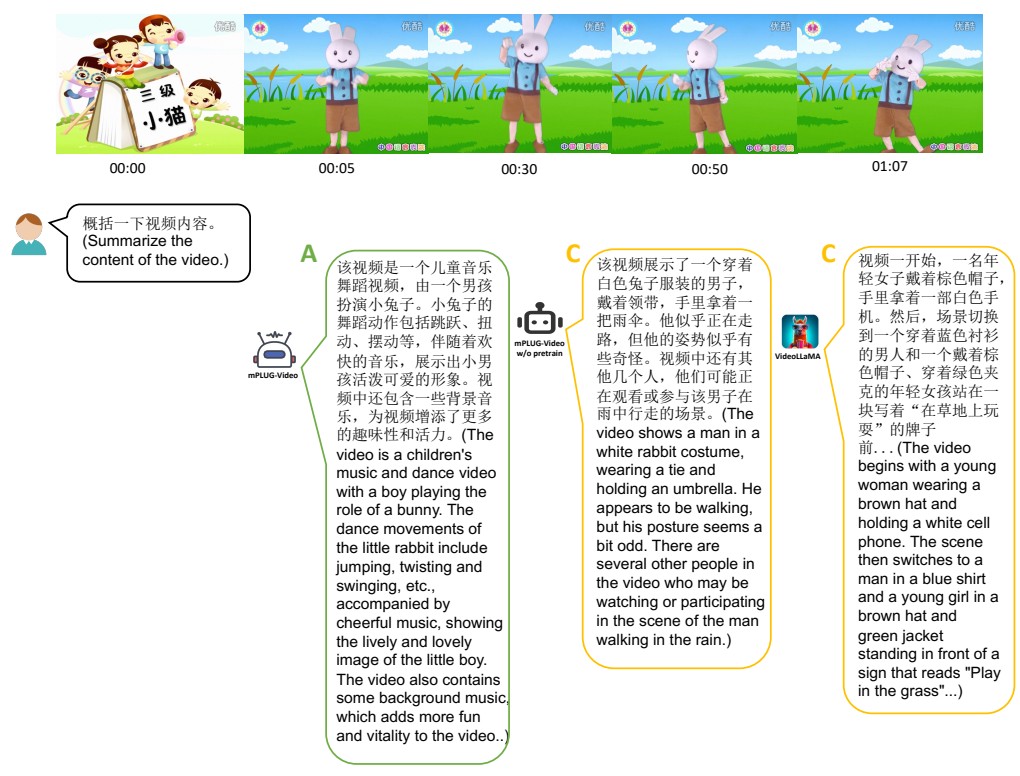

Figure 6: Qualitative results about video captioning.

In this section, we show some qualitative results of zero-shot video instruction understanding to validate the effectiveness of pretraining with Youku-mPLUG. Firstly, as shown in Fig. 6, with pertaining, mPLUG-video can better understand both overall video semantic ('dance video') and detailed vision information, such as 'jumping' and 'twisting'. Besides, Fig. 7 shows that mPLUG-video can better recognize scene texts in videos and give more accurate responses. Therefore, pretraining with Youku-mPLUG is highly likely to enhance the models' ability to comprehend scene text in videos. Furthermore, in Fig. 8, mPLUG-video could accurately point out the key character name 'Ultraman', while the other two models can't. This validates the contribution of pretraining for learning open-domain knowledge.

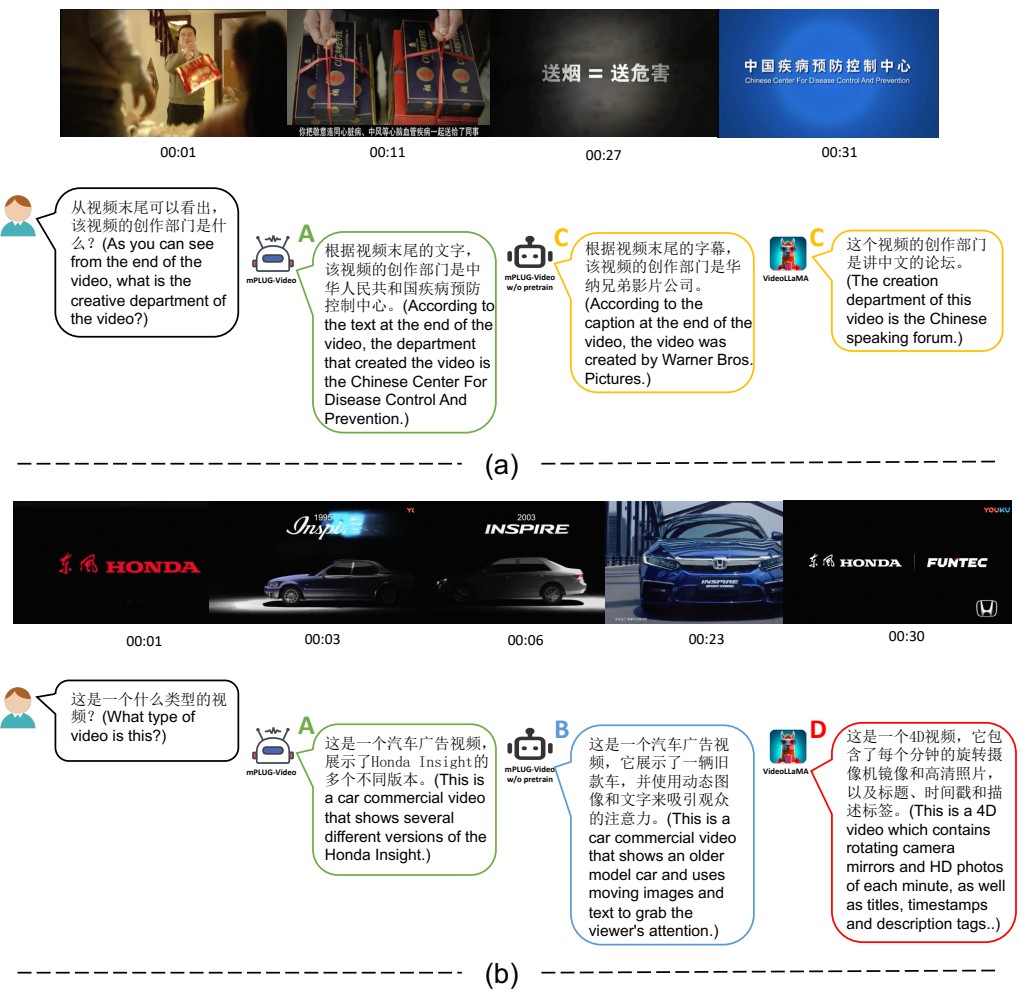

Figure 7: Qualitative results about video scene text understanding.

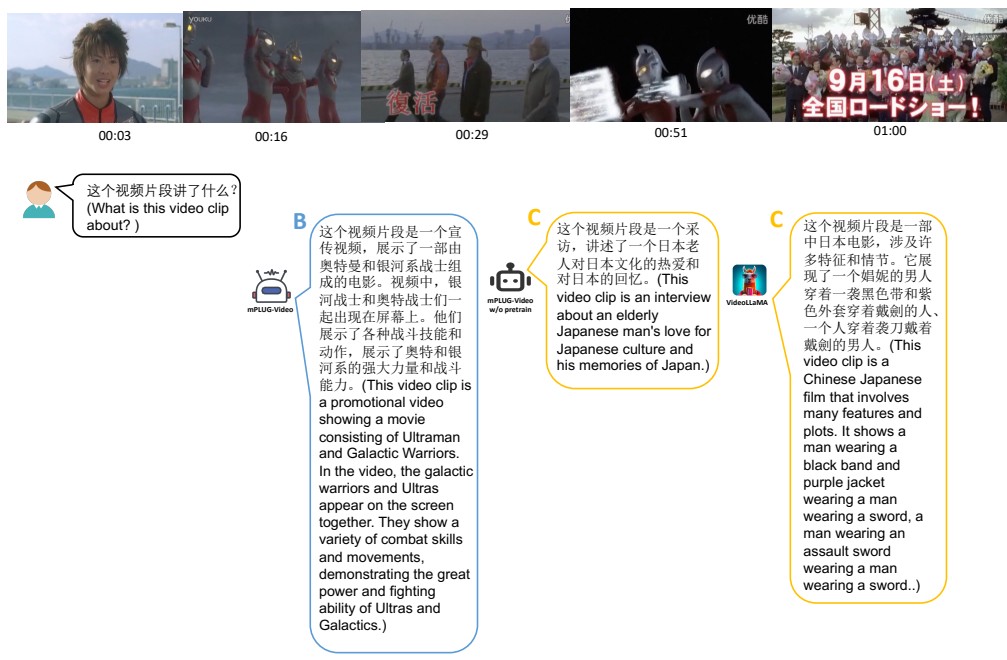

Figure 8: Qualitative results about open-domain knowledge understanding.

## B  LIMITATIONS AND SOCIETAL IMPACTS

The Youku-mPLUG dataset predominantly contains concepts and language expressions that were current at the time of collection. As language evolves alongside human activities, our dataset may not encompass emerging concepts, words, and language expressions in the future. This limitation applies to image data as well, where new visual objects or designs might not be captured. Nevertheless, this issue can be addressed by fine-tuning pre-trained models on up-to-date data. Additionally, our dataset is constructed using corpora from the Chinese Internet, meaning the vocabulary and expressions may largely align with Chinese culture. Furthermore, our dataset lacks very long texts and lengthy videos, potentially limiting the ability of the pre-trained models to understand extensive content such as full-length movies.

## C  HOSTING, MAINTENANCE PLAN, AND LICENSE

Long-term maintenance of Youku-mPLUG and models proposed and evaluated in our paper will be made by the authors. The dataset will be hosted on the Modelscope[4] with Alibaba Cloud as the backend for the downloading service. For stability, we would check the URLs of the dataset regularly and fix those broken videos in time. Our released datasets are provided under the terms of the Creative Commons Attribution-NonCommercial-ShareAlike 4.0 International Public License ("CC BY-NC-SA 4.0"), with the additional terms included herein[5]. When users download or use the datasets from our website, they must agree to the license.

---

[4]https://www.modelscope.cn/
[5]https://creativecommons. org/licenses/by-nc-sa/4.0/legalcode

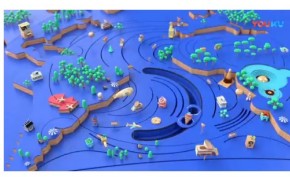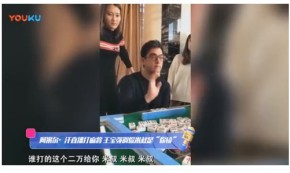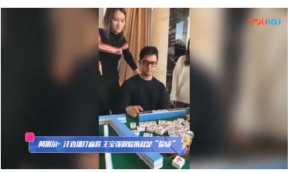

阿米尔汗直播打麻将 王宝强调侃米叔是 "你输" —早班机
Amir Khan plays mahjong live and Wang Baoqiang jokes that "you lose" to Amir Khan - Morning Flight

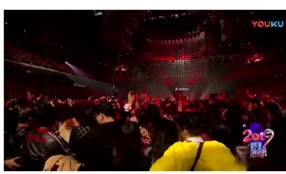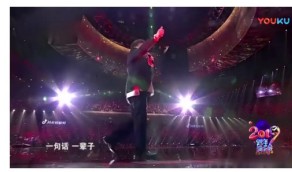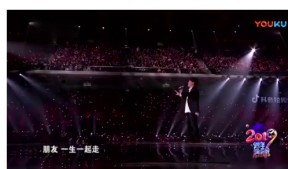

治愈系旋律来袭，周华健经典再现《朋友》，珍惜身边的人吧！
Healing melody strikes, Zhou Huajian classic reproduction of "Friends", cherish the people around you!

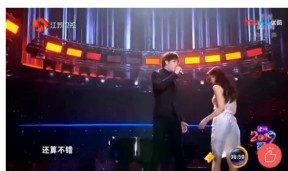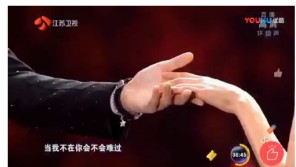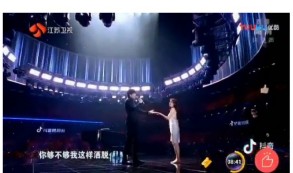

刘宇宁你把手撒开让我来
Liuyuning, let go of your hand and let me do it.

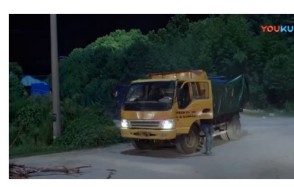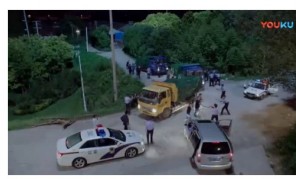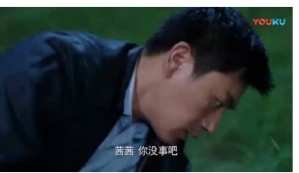

【江河水 第22集】秦昊危机关头舍己为人勇救阚清子尽显真情！
At the crisis point in River of Rivers Episode 22, Qin Hao showed true affection by risking his life to save Kan Qingzi!

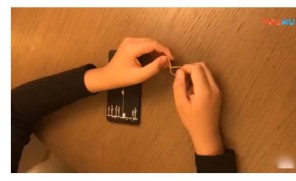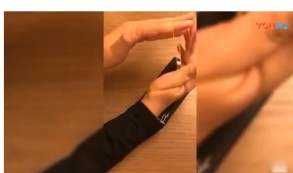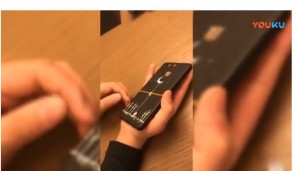

又帅气又简单的橡皮筋魔术1 (校园魔术)
Handsome and Simple Rubber Band Magic 1 (Campus Magic)

Figure 9: Examples in Youku-mPLUG.

