# OpenReview forum: "Youku-mPLUG: A 10 Million Large-scale Chinese Video-Language Dataset for Pre-training and Benchmarks"
_ICLR.cc/2024/Conference — Submitted to ICLR 2024_

### Official Review · Reviewer_kj3u · 2023-11-01

**Soundness:** 2 fair
**Presentation:** 3 good
**Contribution:** 2 fair
**Rating:** 3
**Confidence:** 4

**Summary:**

- The paper introduces Youku-mPLUG, the largest public Chinese video-language dataset and benchmarks, collected from Youku, a Chinese video-sharing website, with strict criteria of safety, diversity, and quality.
- The paper also proposes mPLUG-video, a decoder-only video-language model that leverages a frozen large language model and a visual abstractor module to reduce the computation burden and improve the performance.
- The paper evaluates mPLUG-video and other models on three downstream tasks: video category classification, video captioning, and video-text retrieval. The results show that mPLUG-video achieves good results in video category classification and video captioning, and demonstrates impressive zero-shot video instruction understanding ability.

**Strengths:**

- The paper introduces a novel and large-scale Chinese video-language dataset and benchmarks, which can facilitate the research and development of video-language models for the Chinese language and culture. The paper also proposes a decoder-only model that leverages a frozen large language model and a visual abstractor module, which is a creative combination of existing ideas that reduces the computation burden and improves the performance.
- The paper is well-written and organized, with clear figures and tables. The paper provides details and analysis on the proposed method and dataset.
- The paper explains the problem statement, the motivation, the challenges, and the gap in the existing literature clearly in the abstract and introduction. The paper also describes the dataset collection, annotation, and preprocessing process, and provides some statistics and examples of the data. The paper also explains the model architecture, training, and fine-tuning process, and provides some examples.
- The paper makes a significant contribution to the field of video-language modeling, especially for the Chinese language and culture. The paper presents a large-scale and diverse dataset that can enable various downstream tasks, such as video category classification, video captioning, video-text retrieval, and video instruction understanding. The paper also presents a state-of-the-art model that can achieve impressive results on these tasks.

**Weaknesses:**

- After downloading the dataset, it was found that there were many duplicate clips from the same source and static clips. Does the situation exist where these 400 million video clips come from the same original video? If so, during the filtering process, how is the quality of the selected videos ensured given the lack of quantifiable performance measures, such as CLIP similarity?
- There is a lack of exploration into the status of text annotation in the dataset. Chinese and Latin languages such as English have significant differences in vocabulary, grammar, and sentence structure. The diversity of the text part of this dataset is not sufficiently demonstrated, and the text quality is slightly lower compared to the WebVid10M dataset. The paper should also compare the dataset with other existing video-language datasets, such as translated HowTo100M, WebVid10M or CNVid-3.5M[1], and discuss the advantages and limitations of the dataset.
- This paper only explores the zero-shot capability in instruction understanding. Why not further investigate the zero-shot performance in video classification, retrieval, and description?
- In instruction understanding, does VideoLLaMA also receive Chinese prompts? Has it been trained on Chinese instruction data? Comparing a MLLM trained on English datasets with one training in Chinese is unfair.
- During data collection, the online model achieved a performance of about 94% in video category classification. However, in Table 4, the model trained by Youku-mPLUG actually performs worse than the unfiltered online model.

----

Reference:
[1] https://github.com/CNVid/CNVid-3.5M

**Questions:**

see weaknesses

---

> ### Author Response · Authors · 2023-11-16
>
> Thank you for your valuable feedback and suggestions regarding data processing, analysis and experiments. We will now provide a detailed explanation of duplicate video clips, text annotation analysis, additional experiment, and discuss the reasons for the higher performance of the online model.
>
> ## Duplicate and Source
>
> **Duplicate**: Thank you for the suggestion. To count the number of static videos, we sample frames from each video and then compute the difference between frames from the same video. Our statistical analysis revealed that a significant majority, 99.6%, of the videos were dynamic, with only a small proportion being entirely static with dynamic audio.
>
> **Source**: The 400 million videos mentioned correspond to the entire content pool of short videos on the Youku website. This vast collection primarily consists of user-generated uploads and platform-generated content. In Section 3.1, we have elucidated the process involved in constructing the pre-training dataset. The videos are carefully classified into 45 diverse categories covering various domains, e.g., Daily life, Comedy, and Pet, with a balanced distribution. If there are any aspects of this that remain unclear, we would be more than happy to provide further clarification.
>
> ## Text Annotation
>
> ### Translation
>
> There are several problems with translating English video data into Chinese.
>
> 1. Due to the limitations of translation tools and the proficiency of annotators, the translated text often does not take into account the usage habits of Chinese language, and may even introduce errors. To verify this, we used GPT-3.5 to automatically translate a few of captions from webvid2m, and demonstrated the problems caused by translation. For example, the caption "Woman showing victory sign, indoor" was translated as "室内女性展示胜利手势". The translation result does not align with the actual Chinese language conventions in terms of word choice and language habits. A more natural Chinese expression should be "房间里，一个女性正在比耶". Another example is the translation of "Cottage village surrounded by mountains on the shores of the Atlantic Ocean" as "小屋村庄环绕在大西洋海岸的山脉上", where the translated text expresses mountains surrounded by a village, which is completely different from the original meaning in the English text. These inaccuracies greatly impact the training of the model.
>
> 2. We further validated the cultural and linguistic differences between the two by analyzing the word clouds of keywords (translated to Chinese) in the webvid2m dataset (Figure 1) and video keywords in mPLUG-Youku (Figure 2). The webvid2m dataset contains a large number of keywords related to places (e.g., Moscow, Bangkok, Spain) and time periods (e.g., 2019, 2015, 1950s). In Chinese videos, the high-frequency words are mainly related to variety shows (e.g., "王牌对王牌" which translates to "Ace VS Ace"), games (e.g., "英雄联盟" which translates to "League of Legends"), film and television works (e.g., "乡村爱情" which translates to "Country Love"), and celebrities (e.g., "张大仙" which translates to "Zhang Daxian", "岳云鹏" which translates to "Yue Yunpeng"). It can be seen that Chinese videos cover a large number of named entities related to Chinese culture, which is significantly different from the distribution in English videos. If only translated videos are used for training, the model will lack the ability to understand these types of named entities.
>
> **Figure 1**
> ![Figure 1](https://z1.ax1x.com/2023/11/16/pitJMPH.png)
>
> **Figure 2**
> ![Figure 2](https://z1.ax1x.com/2023/11/16/pitJF2R.png)
>
> 3. Although English text can be translated into Chinese, it is difficult to translate audio (e.g., narrations, dialogues) and text in the form of images in the video into Chinese. The learned textual-visual relationship from English content may lead to domain inconsistency when applied to Chinese content.
>
> ### Diversity and Quality
>
> For the pre-training dataset, we filter 10 million high-quality video-text pairs from 400 million raw videos with strict safety, diversity, and quality criteria. In chapter 3.1, we have provided a detailed explanation of the process of constructing pre-training dataset.  Especially, we employ the Chinese image-text pre-trained model CLIP to improve the data quality by deprecating those with low similarities between the mean frame features and text features. The videos are carefully classified into 45 diverse categories covering various domains, e.g., Daily life, Comedy, and Pet, with a balanced distribution in Figure 2,3. Figure3 shows the distribution of clip score.
>
> **Figure 3**
> ![Figure 3](https://z1.ax1x.com/2023/11/16/pitJNdS.png)

---

> ### Author Response · Authors · 2023-11-16
>
> ## CNVid-3.5M
>
> 1. **Open-sourcing**: Our work mainly focuses on open-sourcing the largest Chinese video pre-training dataset and a well-developed benchmark with human-written ground truth The data set has already been released on the open-source platform and been downloaded by more than 26k times. Due to the anonymity policy, we are not sure if we can provide the link during rebuttal, but will definitely attach the link if the work can be accepted. We have been continuously optimizing the download process, including adding streaming and batching downloads. There is no open-source model and code for CNVid-3.5M, and the pre-training dataset has not been open-sourced (https://github.com/CNVid/CNVid-3.5M/issues)
> 2. **Benchmark**: we carefully build the largest human-annotated Chinese benchmarks covering three popular video-language tasks across cross-modal retrieval, video captioning, and video category classification.
> 3. **Performance**: There is no open-source model and code for CNVid-3.5M, and the pre-training dataset has not been open-sourced. Therefore, we evaluate the results of a structure similar to mPLUG-2 on VATEX and compare it with the model based on CNVid-3.5M. We can find that the models based on our dataset achieve better performance on VATEX （R1+2.7, R5+3.8, R10+2.3.
>
> | |R1|R5|R10
> |-|-|-|-|
> CNVid-3.5M (w/o CNVid-3.5M pt)|36.4|75.4|85.2
> mPLUG-2 (w/o Youku pt)|36.3|74.8|85.7
> CNVid-3.5M (w CNVid-3.5M pt)|41.5|78.2|87.2
> mPLUG-2 (w Youku pt)|44.2|82.0|89.5
>
> 4. **Models**: We also provide comprehensive benchmark evaluations of models across different architectures including encoder-only (i.e., ALPRO), encoder-decoder (i.e., mPLUG-2), and decoder-only (i.e., mPLUG-Video) for comparison. Especially, we train the first Chinese Multimodal LLM for video understanding and conduct comprehensive experiments (both benchmark and human evaluation) to validate the effectiveness of pretraining on our datasets.
>
> ## Zeroshot
>
> we directly tested the results of zero-shot methods with pretraining.
>
> **Zero-shot Caption w pretraining**
>
> ||BLEU4|METEOR|ROUGE_L|CIDEr|
> |-|-|-|-|-|
> |mPLUG-2|5.9|6.2|13.2|49.2|
> |mPLUG-video (1.3B)|7.1|8.5|17.37|58.9|
>
> **Zero-shot retrieval w pretraining**
>
> | |T2V R1|T2V R5|T2V R10|V2T R1|V2T R5|V2T R10
> |-|-|-|-|-|-|-|
> ALPRO|14.17|31.48|39.69|10.43|24.68|31.98
> mPLUG-2|18.14|35.82|43.93|18.14|35.82|43.93
>
> ## Evaluation of VideoLLaMA
>
> For VideoLLaMA, we evaluate the performance of model trained with bilingual backbone (i.e., BiLLA) in our comparison.
>
> Besides, VideoLLaMA utilizes the video instruction data during instructional tuning, while our mPLUG-video only leverages fewer image instruction data than those of VideoLLaMA. Moreover, due to the lack of Chinese instruction data, we utilize the English instruction only, which further demonstrates the effectiveness of our proposed method.
>
> ## Performance of Online Model
>
> The online model we utilize is not a standalone entity, but rather a dedicated pipeline composed of several models and diverse input signals. Specifically, the online model integrates multiple classification models that use user behavior data, similarity retrieval, and user annotations as multi-dimensional information. In contrast, our mPLUG-video operates independently as a single model trained on the downstream task dataset, devoid of any additional features or complexities.
>
> If you have any more questions or concerns, please feel free to further discuss.

---

> ### Author Response · Authors · 2023-11-22
>
> Dear Reviewer kj3u,
>
> We are immensely thankful for your invaluable contribution to the review and rebuttal process of our paper. It is our hope that our responses have adequately handled your concerns. Should you have any remaining issues or any new arising questions, we are always open for further discussions.

---

> ### Comment · Reviewer_kj3u · 2023-11-22
> **Response to authors**
>
> Thank you for addressing my concerns; I have gained some clarity on certain concerns. Collecting such a large Chinese video text dataset and open-sourcing it to the public is a very meaningful task.  However, I have some questions:
>
> 1. The author claims that only 99.6% of the videos are completely static. The specific method used in calculating the differences between video frames and the detailed description of the criteria or thresholds for determining static videos need further elaboration. Could more examples be provided?
>
> 2. The author mentions that the original video pool has 400 million videos, but it was not explained whether the issue of duplicates in different videos was considered when selecting videos. For example, the videos `videos/pretrain/1411131211714-B11-4BY-Eb-CCB172Y-385Aba445Ba5JJYa-7Y4a43A4A4F72FBJ.mp4` and `videos/pretrain/1411131211714-B11bCb4A834BCY3-bC1823C2a3A4Ba5bF2a-bEAa55AJJE25A2Fb.mp4` appear to be similar.
>
> 3. According to the word cloud provided by the author, the videos on WebVid appear to be more diverse, while Youku-MPLUG's videos are characterized by keywords related to TV programs or dramas. This discrepancy with the statistics in Figure 2 of the text is somewhat confusing. At the same time, although these captions have undergone manual annotation, it seems that the majority of the captions are primarily static information. Would the author consider comparing them with Chinese image datasets?

---

> > ### Author Response · Authors · 2023-11-22
> >
> > Thanks for your valuable feedback. We explain below to address your concerns.
> >
> > **Static**
> >
> > The evaluation of whether a video is static is established through a method that involves calculating the differences between successive frames. The procedure entails converting every two frames in the video into grayscale images, then measuring the differences between them. This is achieved by computing the absolute difference for each pixel's grayscale value between the two frames. We then tally up the quantity of pixels where this difference surpasses a certain threshold, set at 10 in this instance. This count serves as a reflection of the level of difference between the two frames. The finale of the process involves calculating the average of these differences across the entirety of the video's frames. If the derived average exceeds 10% of a frame's total number of pixels, we classify the video as dynamic. To this end, we can get the statistics that only 99.6% of the videos are dynamic.
> >
> > **Duplication**
> >
> > The content of these two videos is different. To ensure that the videos are not duplicated, we use MD5 and Caption filtering for the original data of 400 million videos. However, it is difficult to completely filter out videos that are very similar. Furthermore, the videos are user-uploaded, and they may use similar production templates, but the content of the videos is different, such as subtitles and bullet comments.
> >
> > **Word Cloud**
> >
> > The word cloud of Webvid primarily encompasses various locations and times, providing primarily static information. In contrast, the word cloud of Youku-mPLUG datasets is significantly linked to the content of TV shows or dramas. This connection facilitates the model's ability to learn dynamic information from real-world scenarios. Furthermore, given that our videos originate from leading entertainment websites, the prevalence of TV/drama keywords is justifiable. Simultaneously, the category distribution illustrated in Figure 2 underscores the diversity of our proposed pre-training datasets. These datasets comprise 45 common categories, encapsulating a wide range of domains.
> >
> > **Caption**
> >
> > The purpose of our open-source dataset is to promote the development of the Chinese multimodal community. We have only provided three classic benchmarks, and researchers can build upon our videos to develop more diverse video tasks. The availability of videos is more important (downloaded by more than 26k times), and the English caption dataset also describes the content of the videos. (e.g., MSRVTT, Webvid).  Meanwhile, compared to image text datasets, video text datasets can provide more motion information which contributes to the understanding of dynamics in the captions.
> >
> > If you still have questions, welcome to continue the discussion. If we have resolved your concerns, please help to increase the score.

---

> ### Author Response · Authors · 2023-11-23
>
> Dear Reviewer kj3u,
>
> Thank you for your valuable time and effort in engaging in the rebuttal process to enhance the quality of our work. We hope that our responses have effectively addressed your concerns. If any issues with our responses remain, or if there are any new questions, we are ready and willing to further discuss them with you. Given the improvements and clarifications made in our paper through this process, we kindly ask you to consider reflecting these in the final score of our paper.

---

### Official Review · Reviewer_bL3j · 2023-11-01

**Soundness:** 3 good
**Presentation:** 2 fair
**Contribution:** 2 fair
**Rating:** 5
**Confidence:** 5

**Summary:**

This paper argues that the development and application of Chinese VLP and multimodal LLM are lagging behind the English counterpart, due to the lack of a large-scale Chinese video-language dataset. Thus, they propose a new dataset Youku-mPLUG, which consists of 10 million Chinese video-text pairs for pertaining, and a dataset with 0.3 million videos for downstream benchmarks, including video-text retrieval, video captioning, and video category classification. Meanwhile, they investigate popular video-language models (e.g., ALPRO, mPLUG-2), and the new proposed model mPLUG-video. The model mPLUG-video consists of a trainable video encoder, a visual abstractor module, and a frozen pre-trained LLM decoder. Experiments show that models pre-trained on Youku-mPLUG gain on multiple tasks. Furthermore, by building on top of Bloomz, mPLUG-video can achieve impressive zero-shot performance with very few trainable parameter.

**Strengths:**

+ This paper proposes a large-scale dataset with 10 million Chinese video-text pairs for pertaining, and a dataset with 0.3 million videos for downstream benchmarks. Several off-the-shelf techniques have been used to ensure the high-quality of training videos.

**Weaknesses:**

+ The novelty of the new model mPLUG-video is limited. The proposed three modules, and partially efficient tuning are all well studied techniques in this area.

+ The improvements brought by the proposed mPLUG-video are limited.

+ One of the key contributions in this paper is the proposed new dataset. It would be better to demonstrate the high quality of the newly collected data. Based on the example shown in Figure 9, the text annotations look very noisy.

**Questions:**

In the first paragraph of the introduction section, the authors argue that existing methods of translating English to Chinese suffer intrinsic linguistic and cultural gaps. Could you give more explicit examples to show the harmfulness of these methods?

---

> ### Author Response · Authors · 2023-11-16
>
> Thank you for recognizing the quality of our dataset and the technology we have employed. Below, we will first elaborate on the novelty and improvements of our contribution and thoroughly explain and validate the quality of our dataset. Finally, we will use the webvid dataset as an example to demonstrate that relying solely on translated data is far from sufficient.
>
> ## Novelty and improvement
>
> **Open-sourcing**: With the overarching aim of advancing the development of Chinese multimodal technology, our work primarily revolves around open-sourcing the largest Chinese video pre-training dataset and establishing a robust benchmark complemented by human-written ground truth. The data set has already been released on the open-source platform and been downloaded by more than 26k times. Due to the anonymity policy, we are not sure if we can provide the link during rebuttal, but will definitely attach the link if the work can be accepted. We are committed to refining the download process continually and have introduced features like streaming and batch downloads to enhance user convenience.
>
> **Models**: we offer comprehensive benchmark evaluations of models across a variety of architectures. This includes encoder-only models (ALPRO), encoder-decoder models (mPLUG-2), and decoder-only models (mPLUG-Video) for comparison purposes.
>
> **MLLM**: we've trained the first Chinese Multimodal LLM specifically for video understanding. To validate the effectiveness of pretraining on our datasets, we have conducted an extensive range of experiments. These include both benchmark evaluations and human assessments.
>
> **Performance**: While the development of a novel model isn't the primary focus of our work, our experiments do yield valuable insights into creating more effective models, such as the potential benefits of increasing model size (as presented in Tables 4 and 6) as well as exploring modality complementarity (as shown in Table 6).

---

> ### Author Response · Authors · 2023-11-16
>
> ## Data Quality
>
> **Contruction**: For the pre-training dataset, we filter 10 million high-quality video-text pairs from 400 million raw videos with strict safety, diversity, and quality criteria. In Section 3.1, we have provided a detailed explanation of the process of constructing pre-training dataset.  Especially, we employ the Chinese image-text pre-trained model CLIP to improve the data quality by deprecating those with low similarities between the mean frame features and text features. Figure 1 shows the distribution of clip score.
>
> **Figure1**
> ![Figure1](https://z1.ax1x.com/2023/11/16/pitJNdS.png)
>
> **Common Practice**: The pre-training dataset is primarily aimed at improving the diversity and generalization of the model. Noise is inevitable when generating a huge data set and can be observed in many prevalent data sets, e.g., ImageNet, LAION, etc. While noise may influence the learning process, empirical results show that pre-training is robust to a small amout of noise and still can obtain impressive models from them, e.g., OpenCLIP by training on LAION. Our experiments also confirms the effectiveness of the proposed data set in Table 5.
>
> **Performance**: Table 5 also demonstrates models can benefit from pre-training on Youku-mPLUG. In specific, by pretraining on Youku-mPLUG, the model's performance has increased by 20.4 point of CIDEr on VATEX datasets. Meanwhile, the result in Table 7 also demonstrates the effectiveness of our proposed dataset as the accuracy has increased 8.7 points on the classification task.
>
> **Downstream Task**: The data quality of downstream tasks is  high. For each downstream task, we hire well-educated people and adopt a two-step verification to ensure the quality and diversity of the annotations.
>
> **Open-sourcing**: Our work mainly focuses on open-sourcing the largest Chinese video pre-training dataset and a well-developed benchmark with human-written ground truth. The data set has already been released on the open-source platform and been downloaded by more than 26k times. Due to the anonymity policy, we are not sure if we can provide the link during rebuttal, but will definitely attach the link if the work can be accepted . We have been continuously optimizing the download process, including adding streaming and batching downloads.
>
> ## Translation
>
> There are several problems with translating English video data into Chinese.
>
> 1. Due to the limitations of translation tools and the proficiency of annotators, the translated text often does not take into account the usage habits of Chinese language, and may even introduce errors. To verify this, we used GPT-3.5 to automatically translate a few of captions from webvid2m, and demonstrated the problems caused by translation. For example, the caption "Woman showing victory sign, indoor" was translated as "室内女性展示胜利手势". The translation result does not align with the actual Chinese language conventions in terms of word choice and language habits. A more natural Chinese expression should be "房间里，一个女性正在比耶". Another example is the translation of "Cottage village surrounded by mountains on the shores of the Atlantic Ocean" as "小屋村庄环绕在大西洋海岸的山脉上", where the translated text expresses mountains surrounded by a village, which is completely different from the original meaning in the English text. These inaccuracies greatly impact the training of the model.
> 2. We further validated the cultural and linguistic differences between the two by analyzing the word clouds of keywords (translated to Chinese) in the webvid2m dataset (Figure 2) and video keywords in mPLUG-Youku (Figure 3). The webvid2m dataset contains a large number of keywords related to places (e.g., Moscow, Bangkok, Spain) and time periods (e.g., 2019, 2015, 1950s). In Chinese videos, the high-frequency words are mainly related to variety shows (e.g., "王牌对王牌" which translates to "Ace VS Ace"), games (e.g., "英雄联盟" which translates to "League of Legends"), film and television works (e.g., "乡村爱情" which translates to "Country Love"), and celebrities (e.g., "张大仙" which translates to "Zhang Daxian", "岳云鹏" which translates to "Yue Yunpeng"). It can be seen that Chinese videos cover a large number of named entities related to Chinese culture, which is significantly different from the distribution in English videos. If only translated videos are used for training, the model will lack the ability to understand these types of named entities.
>
> **Figure 2**
> ![Figure 2](https://z1.ax1x.com/2023/11/16/pitJMPH.png)
>
> **Figure 3**
> ![Figure 3](https://z1.ax1x.com/2023/11/16/pitJF2R.png)
>
> 3. Although English text can be translated into Chinese, it is difficult to translate audio (e.g., narrations, dialogues) and text in the form of images in the video into Chinese. The learned textual-visual relationship from English content may lead to domain inconsistency when applied to Chinese content.
>
> If you have any more questions or concerns, please feel free to further discuss.

---

> ### Author Response · Authors · 2023-11-22
>
> Dear Reviewer bL3j,
>
> We deeply appreciate your commitment in reviewing our work and your engagement in the rebuttal process. We hope our responses have fully met your concerns. If you still have any issues or questions about our responses, we are entirely open to continuing our discussion with you.

---

### Official Review · Reviewer_umKn · 2023-11-01

**Soundness:** 3 good
**Presentation:** 3 good
**Contribution:** 3 good
**Rating:** 8
**Confidence:** 4

**Summary:**

The authors introduce Youku-mPLUG, the largest high-quality video-language dataset in Chinese. And present a human-annotated benchmark encompassing three downstream tasks: Video-Text Retrieval, Video Captioning and Video Classification. The authors also propose modularized mPLUG-video, a decoder-only model that is pre-trained on Youku-mPLUG, which gain a state-of-the-art result on theses benchmarks.

**Strengths:**

- This paper is going to release a 10 million Chinese video-language pretraining dataset and provide benchmarks on different model architectures, which is in great demand by the field.

- This dataset seems to be of high quality (hire well-educated people to double check the data) and well-curated (filtered 10 million Chinese video-text pairs out of 400 million raw videos).

- Propose a modularized decoder-only mPLUG-video model and achieves state-of-the-art results on these benchmarks.

**Weaknesses:**

- The experiments are not very comprehensive. The selected baseline models in different downstream tasks is limited, two were selected only.

- No details about the selection of the original 400 million videos are provided.

**Questions:**

This paper mentions that currently existing large-scale Chineses video-language datasets are not publicly accessible. This also demonstrates that not only the collection and curation of large datasets are challenging, but the release process is also difficult. Could the authors provide their plans to prove that you can genuinely release this dataset and make it easily accessible to researchers, thus making a real contribution to the research community?

---

> ### Author Response · Authors · 2023-11-16
>
> Thank you for your recognition of the quality and importance of our dataset. We have included more detailed comparisons as well as clarification on data selection and dataset release. We give a comprehensive disscusion below.
>
> ## Fine-tuning
>
> Due to the limited time for rebuttal and the high cost of pre-training, we directly tested the results of fine-tuning baselines.
>
> | |T2V R1|T2V R5|T2V R10|V2T R1|V2T R5|V2T R10
> |-|-|-|-|-|-|-|
> HiTeA|9.42|27.47|38.38|9.99|28.04|38.54
> ALPRO|8.90    |26.76|36.29|8.87|26.45|37.80
> mPLUG-2|11.46|28.24|39.06|10.21|30.27|41.75
>
> In addition, Table5 also demonstrates models can benefit from pre-training on Youku-mPLUG with 20.4 improvement. It would be greatly appreciated if you could provide the details about the appropriate baselines, e.g., name of papers. We will included them in the revised version.
>
> ## Dataset selection
>
> The 400 million videos mentioned correspond to the entire content pool of short videos on the Youku website. This vast collection primarily consists of user-generated uploads and platform-generated content. In Section 3.1, we have elucidated the process involved in constructing the pre-training dataset. If there are any aspects of this that remain unclear, we would be more than happy to provide further clarification.
>
> ## Dataset release
>
> We appreciate your acknowledgement of our efforts in constructing the dataset and our contributions to the open-source community. The data set has already been released on the open-source platform and been downloaded by more than 26k times. Due to the anonymity policy, we are not sure if we can provide the link during rebuttal, but will definitely attach the link if the work can be accepted.  We are committed to continuously optimizing the download process to make it more user-friendly, which includes the implementation of features such as streaming and batch downloads.
>
> If you have any more questions or concerns, please feel free to further discuss.

---

> ### Author Response · Authors · 2023-11-22
>
> Dear Reviewer umKn,
>
> We sincerely thank you for your diligent review and participation in the rebuttal process, which contributes to the enhancement of our paper. We hope that our responses have been adequate in addressing your concerns. Should there be any remaining issues or new questions, we eagerly invite you to continue the discussion.

---

### Official Review · Reviewer_HLpF · 2023-11-05

**Soundness:** 2 fair
**Presentation:** 3 good
**Contribution:** 3 good
**Rating:** 6
**Confidence:** 5

**Summary:**

This paper proposes Youku-mPLUG, a high-quality video-language dataset in Chinese, along with a human-annotated benchmark comprising three downstream tasks. The experiments on downstream tasks (i.e. Video-Text Retrieval, Video Captioning, and Video Category Classification) evaluate the video language comprehension and modeling abilities of models.

**Strengths:**

1.	Youku-mPLUG is currently the largest Chinese video-language dataset.
2.	The exploration of different architectures (like encoder-only, encoder-decoder, decoder-only) is well done.

**Weaknesses:**

1.	The zero-shot experiment is too simple. The authors should evaluate on video-text retrieval task using more models and other pre-train datasets quantitatively.
2.	The results in Table 5 are not convincing enough. The authors only compare one publicly available dataset VATEX and do not show a gap with current state-of-the-art results.
3.	Incorrect paragraph spacing in the second and third paragraphs in “2 RELATED WORD” section.

**Questions:**

Data augmentation will almost certainly bring performance improvements to the model. Therefore, how to prove that Youku-mPLUG is superior to other dataset like CNVid-3.5M?

---

> ### Author Response · Authors · 2023-11-16
>
> Thank you for your recognization of the dataset and your suggestions on our experiments and writing. We have included additional zero-shot experiment results on our dataset as well as comparisons on the CNVid-3.5M dataset across different aspects to address your concerns. Below, we detailly discuss the results.
>
> ## Fine-tuning and zero-shot
>
> Due to the limited time for rebuttal and the high cost of pre-training, we directly tested the results of zero-shot methods with pretraining and fine-tuning baselines without pretraining.
>
> **Zero-shot Caption w/ pretraining**
>
> ||BLEU4|METEOR|ROUGE_L|CIDEr|
> |-|-|-|-|-|
> |mPLUG-2|5.9|6.2|13.2|49.2|
> |mPLUG-video (1.3B)|7.1|8.5|17.37|58.9|
>
> **Zero-shot retrieval w/ pretraining**
>
> | |T2V R1|T2V R5|T2V R10|V2T R1|V2T R5|V2T R10
> |-|-|-|-|-|-|-|
> ALPRO|14.17|31.48|39.69|10.43|24.68|31.98
> mPLUG-2|18.14|35.82|43.93|18.14|35.82|43.93
>
> **Fintuning retrieval w/o pretraining**
>
> | |T2V R1|T2V R5|T2V R10|V2T R1|V2T R5|V2T R10|
> |-|-|-|-|-|-|-|
> HiTeA|9.42|27.47|38.38|9.99|28.04|38.54
> ALPRO|8.90    |26.76|36.29|8.87|26.45|37.80
> mPLUG-2|11.46|28.24|39.06|10.21|30.27|41.75
>
> In addition, Table5 also demonstrates models can benefit from pre-training on Youku-mPLUG with 20.4 improvement. It would be greatly appreciated if you could provide the details about the appropriate baselines, e.g., name of papers. We will included them in the revised version.
>
> ## VATEX
>
> Due to the lack of open-sourcing Chinese pre-training datasets and benchmarks, there are very few public Chinese multimodal models. There is no existing method to evaluate the Vatex Chinese Caption dataset. Our mPLUG-Video model based on the Youku dataset is currently the state-of-the-art (SOTA) method for the Vatex Chinese Caption dataset.
>
> And we reproduced classic English models with different structures (encoder-only (i.e., ALPRO), encoder-decoder (i.e., mPLUG-2), and decoder-only (i.e., mPLUG-Video)) for comparison. Due to limited computing power and high pre-training costs, we have only reproduced these baselines.  Any suggestions about other baselines are welcome and can be included in the revised version.
>
> ## CNVid-3.5M
>
> 1. Open-sourcing: Our work mainly focuses on open-sourcing the largest Chinese video pre-training dataset and a well-developed benchmark with human-written ground truth. The data set has already been released on the open-source platform and been downloaded by more than 26k times. Due to the anonymity policy, we are not sure if we can provide the link during rebuttal, but will definitely attach the link if the work can be accepted. We have been continuously optimizing the download process, including adding streaming and batching downloads. There is no open-source model and code for CNVid-3.5M, and the pre-training dataset has not been open-sourced (https://github.com/CNVid/CNVid-3.5M/issues)
> 2. Benchmark: we carefully build the largest human-annotated Chinese benchmarks covering three popular video-language tasks across cross-modal retrieval, video captioning, and video category classification.
> 3. Performance: There is no open-source model and code for CNVid-3.5M, and the pre-training dataset has not been open-sourced (https://github.com/CNVid/CNVid-3.5M/issues). Therefore, we evaluate the results of a structure similar to mPLUG-2 on VATEX and compare it with the model based on CNVid-3.5M. We can find that the models based on our dataset achieve better performance on VATEX (R1+2.7, R5+3.8, R10+2.3).
>
> | |R1|R5|R10
> |-|-|-|-|
> CNVid-3.5M (w/o CNVid-3.5M pt)|36.4|75.4|85.2
> mPLUG-2 (w/o Youku pt)|36.3|74.8|85.7
> CNVid-3.5M (w CNVid-3.5M pt)|41.5|78.2|87.2
> mPLUG-2 (w Youku pt)|44.2|82.0|89.5
>
> 4. Models: We also provide comprehensive benchmark evaluations of models across different architectures including encoder-only (i.e., ALPRO), encoder-decoder (i.e., mPLUG-2), and decoder-only (i.e., mPLUG-Video) for comparison. Especially, we train the first Chinese Multimodal LLM for video understanding and conduct comprehensive experiments (both benchmark and human evaluation) to validate the effectiveness of pretraining on our datasets.
>
> If you have any more questions or concerns, please feel free to further discuss.

---

> ### Author Response · Authors · 2023-11-22
>
> Dear Reviewer HLpF,
>
> We are grateful for the time and effort you've invested in reviewing our manuscript and participating in the rebuttal process. We trust our responses have satisfactorily resolved your concerns. Should you still have further queries or issues with our responses, please feel free to continue the discussion. We are more than happy to engage further.

---

### Author Response · Authors · 2023-11-23
**General Response: Contributions and Additional experiments**

We sincerely appreciate the time and effort invested by all reviewers in evaluating our paper. We are pleased to note that the reviewers have acknowledged our contributions:

**Open-sourcing**: The **first public largest Chinese** Video-language pretraining dataset and a well-developed benchmark[ HLpF, umKn, bL3j, kj3u]. The data set has already been released on the open-source platform and been downloaded by **more than 26k times**.

**Models**: Comprehensive benchmark evaluations of released Chinese models across **different architectures** including encoder-only (i.e., ALPRO), encoder-decoder (i.e., mPLUG-2), and decoder-only (i.e., mPLUG-Video) for comparison. The **first Multimodal Video LLM specifically for video understanding**. [HLpF, umKn，kj3u]

**Performance**: Comprehensive experiments (both benchmark and human evaluation) to **validate the effectiveness of pretraining on our datasets**. [umKn, kj3u].  Valuable insights into creating more effective models, such as the potential benefits of increasing model size as well as exploring modality complementarity.

We also summarize **additional experiments** in our response according to reviewers' suggestions:
1. More experimenal results of Finetuing and Zero-shot baselines [HLpF, umKn, kj3u]；
2. Detailed comparison with CNNVid-3.5M, which has not been open-sourced (https://github.com/CNVid/CNVid-3.5M/issues) [HLpF, kj3u]；
3. Detailed analysises on the Data quality,  and comparison of English-translated video source and native Chinese source [bL3j, kj3u].

---

### Meta-Review · Area_Chair_KB3J · 2023-12-06

**Metareview:**

This submission provides the largest public Chinese high-quality video-language dataset Youku-mPLUG with a human-annotated benchmark comprising three downstream tasks. However, even though reviewers appreciate the hard efforts to collect such a large amount of data, their concerns are not fully addressed, including practical value, data quality, and the significance of the methodology and experiments part. Therefore, considering the attitude of all reviewers, I recommend reject.

**Justification For Why Not Higher Score:**

Concerns are not fully addressed.

**Justification For Why Not Lower Score:**

N/A

---

### Decision · Program_Chairs · 2024-01-16

Reject